# Extra Virgin Olive Oil Reduces Gut Permeability and Metabolic Endotoxemia in Diabetic Patients

**DOI:** 10.3390/nu14102153

**Published:** 2022-05-21

**Authors:** Simona Bartimoccia, Vittoria Cammisotto, Cristina Nocella, Maria Del Ben, Alessandra D’Amico, Valentina Castellani, Francesco Baratta, Pasquale Pignatelli, Lorenzo Loffredo, Francesco Violi, Roberto Carnevale

**Affiliations:** 1Department of Medical-Surgical Sciences and Biotechnologies, Sapienza University of Rome, Corso della Repubblica 79, 40100 Latina, Italy; simona.bartimoccia@uniroma1.it (S.B.); roberto.carnevale@uniroma1.it (R.C.); 2Department of Clinical Internal, Anaesthesiological and Cardiovascular Sciences, Sapienza University of Rome, Viale del Policlinico, 155, 00161 Rome, Italy; vittoria.cammisotto@uniroma1.it (V.C.); cristina.nocella@uniroma1.it (C.N.); maria.delben@uniroma1.it (M.D.B.); farmcesco.baratta@uniroma1.it (F.B.); pasquale.pignatelli@uniroma1.it (P.P.); lorenzo.loffredo@uniroma1.it (L.L.); 3Department of Movement, Human and Health Sciences, University of Rome “Foro Italico”, 00135 Rome, Italy; a.damico@studenti.uniroma4.it; 4Department of General Surgery and Surgical Speciality Paride Stefanini, Sapienza University of Rome, Viale del Policlinico, 155, 00161 Rome, Italy; valentina.castellani@uniroma1.it; 5Mediterranea Cardiocentro-Napoli, Via Orazio, 2, 80122 Naples, Italy

**Keywords:** lipopolysaccharides (LPS), gut permeability, zonulin, Glucagon-like peptide1(GLP1), extra virgin olive oil (EVOO), oleuropein, Impaired fasting glucose (IFG)

## Abstract

Background: Extra virgin olive oil (EVOO) improves post-prandial glycemia, but the underlying mechanism has not been fully elucidated. We tested the hypothesis that EVOO improves post-prandial glycemia by reducing gut permeability-derived low-grade endotoxemia. Methods: Serum levels of lipopolysaccharides (LPS), zonulin, a marker of gut permeability, glucose, insulin and glucagon-like peptide 1 (GLP1) were measured in 20 patients with impaired fasting glucose (IFG) and 20 healthy subjects (HS) matched for sex and age. The same variables were measured in IFG patients (*n* = 20) and HS (*n* = 20) before and after a Mediterranean diet with 10 g EVOO added or not (*n* = 20) or in IFG patients (*n* = 20) before and after intake of 40 g chocolate with EVOO added or not. Results: Compared to HS, IFG had higher levels of LPS and zonulin. In HS, meal intake was associated with a significant increase of blood glucose, insulin, and GLP1 with no changes of blood LPS and zonulin. Two hours after a meal intake containing EVOO, IFG patients showed a less significant increase of blood glucose, a more marked increase of blood insulin and GLP1 and a significant reduction of LPS and zonulin compared to IFG patients not given EVOO. Correlation analysis showed that LPS directly correlated with blood glucose and zonulin and inversely with blood insulin. Similar findings were detected in IFG patients given a chocolate added or without EVOO. Conclusion: Addition of EVOO to a Mediterranean diet or chocolate improves gut permeability and low-grade endotoxemia.

## 1. Introduction

Circulating lipopolysaccharides (LPS), a membrane component of Gram-negative gut microbiota, may be detectable in human circulation in concentrations ranging from as low as 1 pg/mL to as high as 200 pg/mL [1]. In non-septic conditions, low-grade endotoxemia by LPS may be detectable in patients at risk of cardiovascular events such as patients with obesity or type 2 diabetes mellitus (T2DM) or in patients with clinically overt vascular disease [2,3]. In T2DM, the term “metabolic endotoxemia” has been coined to describe the association between low-grade endotoxemia and metabolic changes such as insulin resistance, hyperglycemia and lipid metabolism changes favoring obesity [4]. Metabolic endotoxemia has been suggested to occur as a consequence of enhanced gut permeability, which depends on transcellular or paracellular LPS translocation into systemic circulation [5]. Thus, modulation of gut permeability may have positive effects on glycemic control in T2DM. We and others have previously reported that in diabetic patients, intake of EVOO or its component oleuropein lowered post-prandial LPS and glycaemia, an effect of potentially clinical relevance, as post-prandial glycemia may increase the risk of cardiovascular disease [6,7,8,9]. However, we did not investigate the relationship, if any, between gut permeability and glycaemic control and if EVOO could improve the glycaemic profile by modulating gut permeability. EVOO is composed of 98–98.5% triglycerides [6] and 1.5–2% of minor components such as sterols [7], fatty alcohols [7], waxes [8], phenols, tocopherols, and carotenoids [9]; furthermore, EVOO is rich in polyphenols and vitamin E [10], which could exert a beneficial effect against cardiovascular events [11,12] and cancer [13], even if further randomized clinical studies are necessary to confirm these hypotheses. The antioxidant property of EVOO could turn useful in the context of gut permeability, which is, in fact, enhanced by oxidative stress-mediated zonulin activation and eventually down-regulation of adhesive proteins such as tight junctions [14]. Therefore, we hypothesized that acute administration of EVOO could improve metabolic endotoxemia, i.e., metabolic profile and endotoxemia by LSP, by affecting gut permeability as assessed by serum zonulin, an indirect marker of gut permeability [15]. To explore this issue, we compared the parallel changes of LPS, glycaemia, and serum zonulin in patients with impaired fasting glucose (IFG) and healthy subjects (HS).

## 2. Materials and Methods

### 2.1. Study Design and Participants

We performed a cross-sectional analysis of variables exploring gut permeability, low-grade endotoxemia by LPS and metabolic profile in HS (*n* = 20, 11 males, 9 females, aged 47.05 ± 6.41 years) and in patients with IFG (*n* = 20, 12 males, 8 females, aged 51.54 ± 8.02) (study 1). According to the American Diabetes Association guidelines [16], IFG was defined as a fasting blood sugar glucose concentration ≥100 and <126 mg/dL.

Additionally, we performed a secondary analysis of two previous reports in which EVOO was added to or not from a Mediterranean-type meal (studies 2 and 3) or to 40 g chocolate (study 4) [17,18,19].

Study 2: In this study, we analyzed 20 healthy subjects (HS; 11 males and 9 females) who received a typical Mediterranean lunch including or not 10 g of EVOO in a cross-over design; there was an interval of 30 days between the two phases of the study. Clinical and demographic characteristics of HS are reported in Table 1.

EVOO was provided by monoculture (Itri area, Latina) and its chemical characterization is reported in Appendix A.

Study 3: In this study, we performed an interventional study in 20 IFG patients (12 males and 8 females); clinical and demographic characteristics of IFG are reported in Table 1. We randomized 20 IFG to receive a typical Mediterranean lunch including or not 10 g of EVOO in a cross-over design; there was an interval of at least seven days between the two phases of the study. The Mediterranean-type lunch composition is reported in Appendix A. Briefly, the Mediterranean-type meal for both studies 2 and 3 consisted of pasta (g 100), chicken breast (150 g), salad (80 g), bread (80 g), and apple (200 g) for a total of 894 calories; blood analyses were performed before and 2 h after meal intake.

Study 4: We analyzed the acute effect of chocolate intake in a single blind, crossover study including IFG patients (*n* = 20), randomized to receive 40 g of chocolate spread mixed or not with EVOO. Blood analyses were carried out before and 2 h after the intake of 40 g chocolate (18.5% hazelnuts, 40% of cocoa, sugar, whole milk powder to reach 27% vitamin E over 100 g carbohydrates); the control chocolate had the same characteristics but without EVOO addition. Every blood determination was performed blind. None of the participants were receiving antioxidants supplements, statins, or dietary restrictions prior to the study. The study conformed to the ethical guidelines of the 1975 Declaration of Helsinki and was approved by the Ethical Committee of Sapienza University (Rif. N° 509/2016).

### 2.2. Serum Glucose and Insulin Assay

Glucose and insulin were analyzed in serum samples by a commercial ELISA Kit (Arbor Assay (Ann Arbor, MI, USA) and DRG International (Springfield, NJ, USA)). Glucose values were expressed as mg/dl and intra-assay and inter-assay coefficients of variation were 6% and 9%, respectively. Insulin values were expressed as μU/mL and intra-assay and inter-assay coefficients of variation were 2.2% and 4.5%, respectively.

### 2.3. Serum Glucagon Like Peptide-1 (GLP1) Assay

A commercial ELISA Kit (DRG International) was used for the quantitative determination of bioactive GLP1 (7e36) and (9e36) levels in serum. GLP-1 values are expressed as pmol/L and both intra- and inter-assay coefficients of variation were <10%.

### 2.4. Serum LPS Assay

LPS were measured in serum using a commercial ELISA kit (Cusabio, Wuhan, China). The standards and samples were plated for 2 h at room temperature into a micro-plate pre-coated with the antibody specific for LPS. After incubation, samples were read at 450 nm. Values were expressed as pg/mL; intra-assay and inter-assay coefficients of variation were <10%.

### 2.5. Serum Zonulin Assay

Zonulin concentration was detected in serum by a commercial ELISA kit (Elabscience Houston, TX, USA). Values were expressed as ng/mL; both intra-assay and inter-assay coefficients of variation were within 10%.

### 2.6. Statistical Methods

Categorical variables are reported as percentage and continuous variables as means ± SD unless otherwise indicated. Independence of categorical variables was tested by the chi-square test. Comparisons between groups were analyzed by Student’s *t*-test and were replicated as appropriate with nonparametric tests (Kolmogorov–Smirnov (z) test in case of non-homogeneous variances as verified by Levene’s test).

The cross-over study data were analysed for the assessment of treatment and period effects, by performing a split-plot ANOVA with one between-subject factor (treatment sequence) and two within-subject factors (pre- vs. post-treatment). The full model was considered, allowing for the assessment of all main effects and interactions. Pairwise comparisons were corrected by the Bonferroni test; results were expressed as means ± SD. Bivariate analysis was calculated by the Spearman rank correlation test. A value of *p* < 0.05 was considered statistically significant. All analyses were carried out with GraphPad Prism9.1.0 and IBM SPSS 25.0.

## 3. Results

Clinical characteristics of the population are reported in Table 1.

At baseline, patients with IFG had higher levels of blood glucose, insulin and GLP-1 compared to HS (Figure 1a–c); furthermore, patients with IFG had higher blood levels of LPS and zonulin compared to HS (Figure 1d,e).

As previously reported [17] (Study 2), in HS meal intake was associated with a significant increase of blood glycaemia, insulin, and GLP-1 (Figure 2a–c). A trend of an increase of blood LPS and zonulin was detected but the difference was not significant (Figure 2d,e). LPS did not correlate with blood glucose, insulin, and zonulin; zonulin was not correlated with GLP-1 (Figure 2f–i).

As previously reported [18] (Study 3), two hours after a meal intake containing EVOO, IFG patients showed a less significant increase of blood glucose and a more marked increase of blood insulin and GLP-1 (Figure 3a–c) compared to IFG not given EVOO. Additionally, compared to IFG patients not given EVOO, a significant reduction of blood LPS and zonulin was detected (Figure 3d,e). Correlation analysis showed that LPS directly correlated with blood glucose and zonulin and inversely with blood insulin; blood zonulin inversely correlated with GLP-1 (Figure 3f–i).

Similar modifications of metabolic profile, blood LPS, and zonulin as well as correlation analyses were observed in IFG patients (*n* = 20) taking chocolate with EVOO added compared to controls (Figure 4a–i).

## 4. Discussion

The results of the present study demonstrate that, in IFG patients, EVOO counteracts LPS increases detected after meal or chocolate intake coincidentally with zonulin lowering, suggesting that this beneficial effect may be attributed to improvement of gut permeability.

Previous studies reported that patients with metabolic diseases such as those with T2DM or obesity display low-grade endotoxemia consequent to changes of proteins implicated in intestinal epithelial cell permeability, such as the tight junction (TJ) proteins ZO-1 or occludin [20]. In accordance with these reports, we found elevated levels of LPS in patients with IFG compared to controls coincidentally with zonulin elevation, which increases gut permeability by disassembling the TJ proteins [21]; this finding corroborates the hypothesis that T2DM is associated with metabolic endotoxemia consequent to changes of gut permeability [5,22]. Low-grade endotoxemia has a negative effect on metabolic profile as documented by an experimental study showing that LPS infusion increases glycaemia by negatively affecting insulin activity [5]; in accordance with this, we found that in IFG, but not in HS, LPS correlated directly with glycaemia and inversely with insulin. In two previous studies [21,23], we reported that a meal intake is associated with low-grade endotoxemia in IFG but not in HS but did not investigate the interplay between low-grade endotoxemia and the post-prandial metabolic profile or the relationship between post-prandial LPS and gut permeability. While the inverse relationship between low-grade endotoxemia and metabolic profile confirms previous data in T2DM [4,21], here we report that in patients with IFG intake of a Mediterranean-type meal or chocolate is associated with a coincident increase of LPS and zonulin, suggesting that changes of gut permeability may account for post-prandial low-grade endotoxemia. This finding would suggest, therefore, that in addition to LPS translocation to systemic circulation via chylomicron biosynthesis [24], post-prandial low-grade endotoxemia occurs via a change of gut permeability.

In two separate studies we also reported that EVOO lowers low-grade endotoxemia, but the underlying mechanism was not investigated [21,23]; furthermore, we did not analyze if the positive effect of EVOO on the metabolic profile could be related to a change of low-grade endotoxemia. The experiments conducted that adding EVOO to a Mediterranean-type meal or to chocolate showed a coincident decrease of LPS and zonulin, indicating that EVOO has a positive effect on gut permeability. In order to investigate the underlying mechanism, we focused on GLP-1, which is increased by EVOO administration [9,16,17], and is suggested to be implicated in the up-regulation of TJ proteins [23]; of note, we found a significant relationship between post-prandial increase of GLP-1 and the decrease of LPS and zonulin, suggesting GLP-1-mediated up-regulation of TJ proteins as a potential mechanism accounting for EVOO-related improvement of gut permeability. These data extend our previous report in this field, indicating that the positive effect of EVOO on metabolic profile may be related to changes of gut permeability and in turn to reduction of low-grade endotoxemia. In accordance with this, a coincident post-prandial LPS and glycaemia lowering along with an inverse relationship between LPS and insulin were detected.

The study has implications and limitations. We must acknowledge that serum zonulin is an indirect marker of gut permeability, and thereby further study is necessary to assess the relationship between EVOO intake and gut permeability. We cannot exclude that zonulin may also be implicated in LPS translocation to systemic circulation via mechanisms not related to TJ disassembling. Additionally, we cannot exclude that EVOO may lower circulating LPS by interfering with chylomicron biosynthesis. The study has been carried out in a single center and in Caucasian patients; further study is, therefore, necessary to support the present findings. The study has been conducted in acute conditions, and further study should be carried out to assess the effect of chronic EVOO on gut permeability and low-grade endotoxemia. Another limitation of the study is represented by the lack of further reference points (beyond 2 h from baseline values) that could have better described the post-prandial time course of the variables (glucose, insulin, GLP1, LPS, and zonulin).

## 5. Conclusions

The study reports that EVOO added to a Mediterranean-type diet or chocolate blunts the increase of glycaemia by interfering with markers of gut permeability and low-grade endotoxemia in IFG. Further study is necessary to assess if a similar effect can be detected by chronic administration of EVOO.

## Figures and Tables

**Figure 1 nutrients-14-02153-f001:**
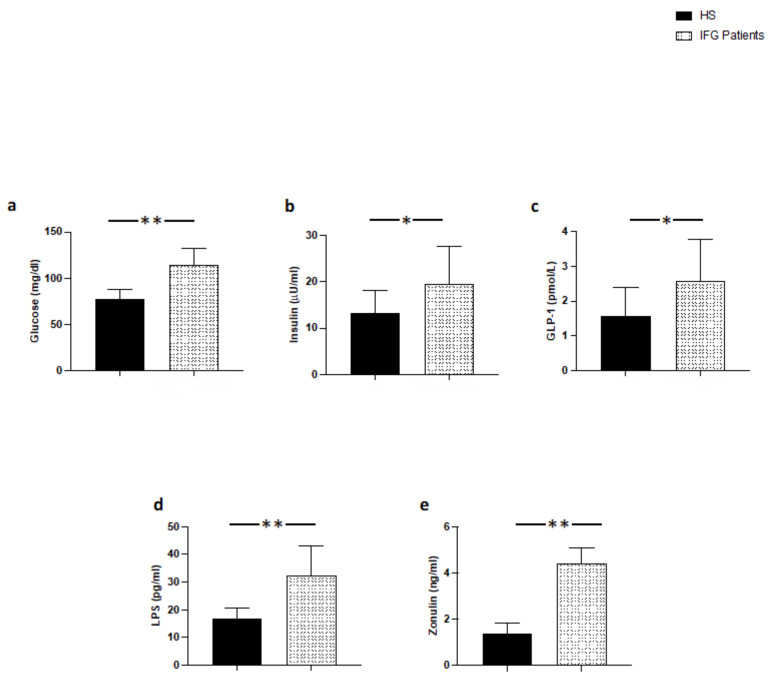
Serum levels of (**a**) glucose, (**b**) insulin, (**c**) GLP-1, (**d**) LPS, and (**e**) zonulin in patients with IFG (*n* = 20) and control subjects (*n* = 20); ** *p* < 0.001; * *p* < 0.01.

**Figure 2 nutrients-14-02153-f002:**
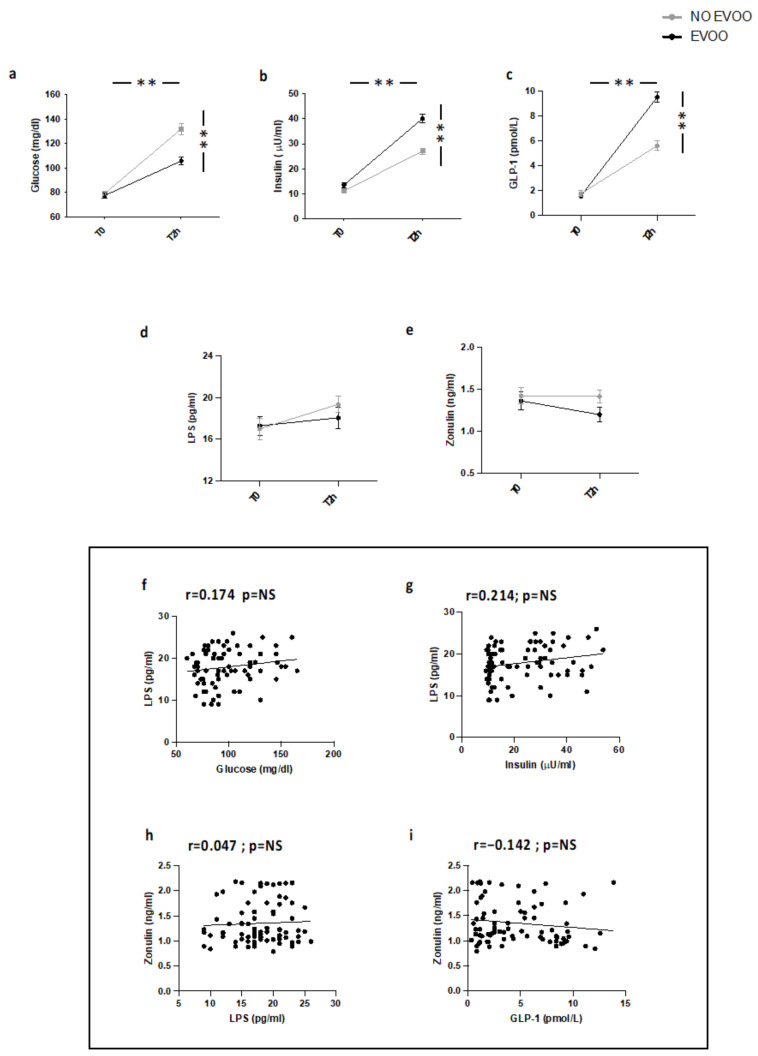
Serum levels of (**a**) glucose, (**b**) insulin, (**c**) GLP-1, (**d**) LPS, and (**e**) zonulin before (T0) and 2 h after a meal with (black line) or without (grey line) extra-virgin olive oil (EVOO) in HS (*n* = 20). Correlations of circulating LPS levels with (**f**) glucose, (**g**) insulin, and (**h**) zonulin in HS. Correlations of circulating Zonulin levels with (**i**) GLP-1 in HS. ** *p* < 0.001.

**Figure 3 nutrients-14-02153-f003:**
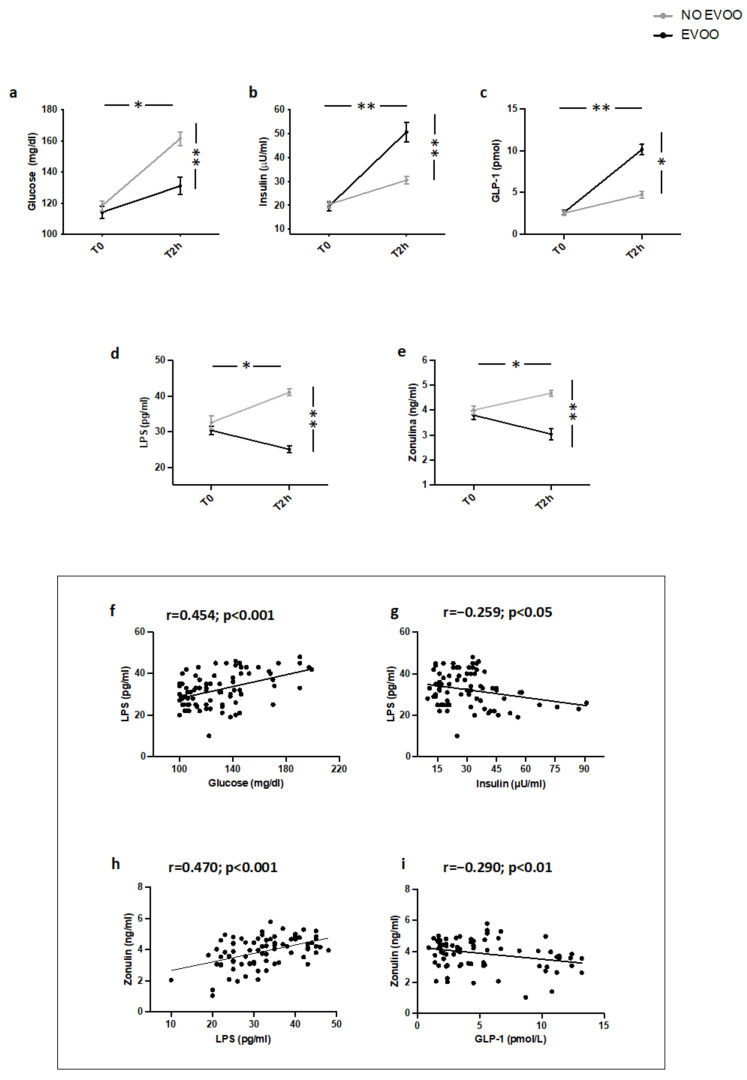
Serum levels of (**a**) glucose, (**b**) insulin, (**c**) GLP-1, (**d**) LPS, and (**e**) zonulin before (T0) and 2 h after (T2h) a meal with (black line) or without (grey line) extra-virgin olive oil (EVOO) in IFG patients (*n* = 20). Correlations of circulating LPS levels with (**f**) glucose, (**g**) insulin, and (**h**) zonulin in IFG. Correlations of circulating Zonulin levels with (**i**) GLP-1 in IFG. ** *p* < 0.001, * *p* < 0.05.

**Figure 4 nutrients-14-02153-f004:**
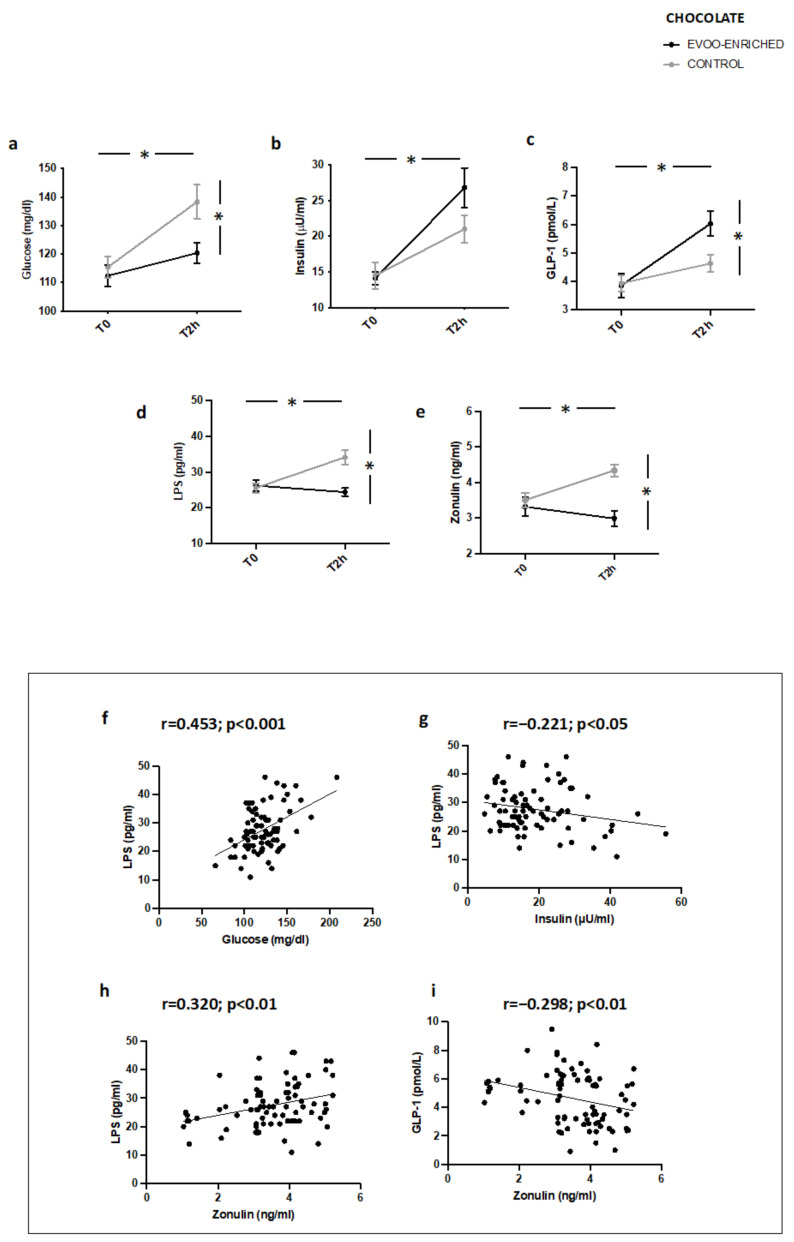
Serum levels of (**a**) glucose, (**b**) insulin, (**c**) GLP-1, (**d**) LPS, and (**e**) zonulin before (T0) and after 2 h (T2h) of oleuropein-enriched chocolate (black line) or control chocolate (grey line) in IFG patients (*n* = 20). Correlations of circulating LPS levels with (**f**) glucose, (**g**) insulin, and (**h**) zonulin in patients with IFG. Correlations of circulating zonulin levels with (**i**) GLP-1 in patients with * *p* < 0.001.

**Table 1 nutrients-14-02153-t001:** Clinical characteristics of the study population.

	HS (*n* = 20)	IFG (*n* = 20)	*p* Value
Age (years)	47.05 ± 6.41	51.54 ± 8.02	0.057
Males *n* (%)	11 (55)	12 (60)	0.757
BMI (kg/m^2^)	27.50 ± 3.91	28.79 ± 3.52	0.270
Systolic BP (mmHg)	121.40 ± 8.52	127 ± 11.29	0.085
Diastolic BP (mmHg)	76.85 ± 4.29	80.00 ± 6.28	0.072
Smokers *n* (%)	4 (20)	2 (10)	0.388

## Data Availability

Not applicable.

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
