# Peer review of "Extra Virgin Olive Oil Reduces Gut Permeability and Metabolic Endotoxemia in Diabetic Patients"

_nutrients, 2022, doi:10.3390/nu14102153_

Round 1

Reviewer 1 Report

The authors have provided more clarity to the manuscript, which noticeably has improved it's coherence. The paper is interesting and the results appear reproducible; the major limitation remains that the biomarker data is not followed beyond 2h, which is a shame. However, the authors have acknowledged this as a clear limitation.  

Author Response

The authors have provided more clarity to the manuscript, which noticeably has improved it's coherence. The paper is interesting, and the results appear reproducible; the major limitation remains that the biomarker data is not followed beyond 2h, which is a shame. However, the authors have acknowledged this as a clear limitation.  

Answer: Thank you very much for this comment

Reviewer 2 Report

Dear Authors, here I have listed my comments.

The manuscript has to be presented as per instructions for authors of Nutrients (MDPI);

  1. The introduction section is very poor and has to be extended in relation with the olive oil composition and in relation with the effect of EVOO on human health;
  2. Introduction section, the argument of your study has to be better presented and the state of the art has to be better discussed; explain that the olive oil composition is influenced by many factors. Please, find, read and discuss some proper reference about pre and post-harvest factors influencing olive oil quality;
  3. Introduction section, your study is focused on extra virgin olive oil but nothing you have written about the extra virgin olive oil composition. Please, explain that EVOO is composed by: 98-98.5% triglycerides [1] and 1.5-2% minor components such as: sterols [2]; fatty alcohols [3]; waxes [4]; phenols, tocopherols, carotenoids [5]. Please, find, read and discuss the references I have listed and include each proper reference after each class of compounds and do not cumulate the references at the end of the sentence:

[1] Influence of cultivar and harvest year on triglyceride composition of olive oils produced in Calabria (Southern Italy).

European Journal of Lipid Science and Technology, 115 (8) 928-934 (2013)

DOI: 10.1002/ejlt.201200390.

[2] Sterol composition of virgin olive oil of forty-three olive cultivars from the World Collection Olive Germplasm Bank of Cordoba.

Journal of The science of food and Agriculture, 96, (12) 4143-4150 (2016). https://doi.org/10.1002/jsfa.7616

[3] The effects of cultivar and harvest year on the fatty alcohol composition of olive oils from Southwest Calabria (Italy).
Grasas y Aceites 65: e011 (2014).

http://dx.doi.org/10.3989/gya.073913

[4] Influence of harvest year and cultivar on wax composition of olive oils.

Eur. J. Lipid Sci. Technol. 115 (5) 549-555 (2013)

DOI: 10.1002/ejlt.201200235.

[5] Effect of crop season on the quality and composition of extra virgin olive oils from Greek and Spanish varieties grown in the Oriental region of Morocco.

Emirates Journal of Food and Agriculture. 2018. 30(7): 549-562

doi: 10.9755/ejfa.2018.v30.i7.1738

  1. The materials and methods section requires some corrections
  2. 2.1 sub-section, line 65, separate 6.41 by years. Also lines 65-66, please be consistent in the whole manuscript and use the same spacing before and after ±;
  3. 2.1 sub-section, line 70: separate study by 4;
  4. 2.1 sub-section, lines 82-83 and in the whole manuscript, the abbreviation for grams is g and not gr;
  5. 2.1 sub-section, and in the whole manuscript, please use the bold only if required by the instructions for authors of Nutrients;
  6. 2.1 sub-section: briefly, which type of pasta?
  7. 2.1 sub-section: briefly, which type of salad?
  8. 2.1 sub-section: briefly, which type of bread?
  9. 2.1 sub-section, line 88: which type of cocoa? Which type of sugar?
  10. 2.1 sub-section. They exist many types of EVOO and not the EVOO. Please, detail the type of EVOO you have used in your experiment: geographical area of production; labelled as EVOO or have you conducted analyses? Year of production and year of your experiment; monocultivar or blended (if possible detail the cultivar/s);
  11. 2.1 sub-section, the authors indicate references 6, 10, 11 in which is reported a part of the experimental method. Yes, I understand, but an International reader cannot find and read 4 papers (3+ 1 actual) to understand the experimental design. Please, describe the experimental design in this manuscript (briefly if you want but insert the relevant information, mainly he ones I have listed in my comments;
  12. Table 1, please use the template to prepare a table;
  13. Table 1 and in the whole manuscript, when you have indicated the significance, sometime you have used P and sometime p. Please, be consistent. Anyway, here you have written P in the head of the column and it is not necessary to repeat for each line;
  14. Figure 2h, and in the whole manuscript, please, separate decimals by a dot and not by a comma;
  15. Figure 3, and in the whole manuscript, please, separate decimals by a dot and not by a comma;
  16. Figure 4, and in the whole manuscript, please, separate decimals by a dot and not by a comma;
  17. Results have to be better presented and argued;
  18. Discussion has to be widely extended also in relation with other studies conducted on this argument in light of the results obtained by the Authors. This section is too poor;
  19. References section is not arranged as required by Nutrients. For example, ref 6 (et al);
  20. References section: sometime you have written the title of the paper in small letters (ref 11) and sometime in capital letters (ref 12), please, be consistent and use some recently published paper as a template. Small or capital letters?
  21. References section. After the last page of the reference you have to write a dot and not a comma, please see you ref 13 and the whole section;
  22. Please, write in blue color or evidence differently the corrections you will do also in the references section.

In my opinion, a major revision is necessary.

Regards.

Author Response

1.The introduction section is very poor and has to be extended in relation with the olive oil composition and in relation with the effect of EVOO on human health.

Answer: According to your suggestion we added the EVOO composition focusing also on its antioxidant property, that is peculiar in the context of our research. As suggested, we also added some data on the effect of EVOO in human health even if studies in human are inconclusive (apart from the PREDIMED study) and need to be confirmed.

See correction at pag. 2 lines 58-63

 “EVOO is composed of 98-98.5% triglycerides [6] and 1.5-2% of  minor components such as sterols [7]; fatty alcohols [7] ; waxes [8]; phenols, tocopherols, carotenoids [9]; furthermore, EVOO is  rich of polyphenols and vitamin E [10] which could exert a beneficial effect against cardiovascular events [11,12] and cancer [13], even  if further  randomized clinical studies are necessary to confirm these  hypotheses.”

2. Introduction section, the argument of your study has to be better presented and the state of the art has to be better discussed; explain that the olive oil composition is influenced by many factors. Please, find, read and discuss some proper reference about pre and post-harvest factors influencing olive oil quality;

Answer: We reported the composition of EVOO in the present study separately (see above). The fact that the quality of EVOO may be influenced by many factors is out of the scoop of this manuscript.

3. Introduction section, your study is focused on extra virgin olive oil but nothing you have written about the extra virgin olive oil composition. Please, explain that EVOO is composed by: 98-98.5% triglycerides [1] and 1.5-2% minor components such as: sterols [2]; fatty alcohols [3]; waxes [4]; phenols, tocopherols, carotenoids [5]. Please, find, read and discuss the references I have listed and include each proper reference after each class of compounds and do not cumulate the references at the end of the sentence:

[1] Influence of cultivar and harvest year on triglyceride composition of olive oils produced in Calabria (Southern Italy). European Journal of Lipid Science and Technology, 115 (8) 928-934 (2013)DOI: 10.1002/ejlt.201200390.

[2] Sterol composition of virgin olive oil of forty-three olive cultivars from the World Collection Olive Germplasm Bank of Cordoba. Journal of The science of food and Agriculture, 96, (12) 4143-4150 (2016). https://doi.org/10.1002/jsfa.7616

[3] The effects of cultivar and harvest year on the fatty alcohol composition of olive oils from Southwest Calabria (Italy). Grasas y Aceites 65: e011 (2014). http://dx.doi.org/10.3989/gya.073913

[4] Influence of harvest year and cultivar on wax composition of olive oils. Eur. J. Lipid Sci. Technol. 115 (5) 549-555 (2013) DOI: 10.1002/ejlt.201200235.

[5] Effect of crop season on the quality and composition of extra virgin olive oils from Greek and Spanish varieties grown in the Oriental region of Morocco. Emirates Journal of Food and Agriculture. 2018. 30(7): 549-562 doi: 10.9755/ejfa.2018.v30.i7.1738

Answer: We quoted but did not discuss the above references as our study is essentially focused on the EVOO antioxidant property. However, your input is of interest in the context of the minimum antioxidant property necessary to exert an antioxidant property by EVOO in vivo. In other words, it is still unclear if each EVOO exerts an antioxidant property or if there is a dose-response curve.

4. 2.1 sub-section, line 65, separate 6.41 by years. Also, lines 65-66, please be consistent in the whole manuscript and use the same spacing before and after ±;

Answer: Amended as suggested.

5. 2.1 sub-section, line 70: separate study by 4;

Answer: Amended as suggested.

6. 2.1 sub-section, lines 82-83 and in the whole manuscript, the abbreviation for grams is g and not gr;

Answer: Amended as suggested.

7. 2.1 sub-section, and in the whole manuscript, please use the bold only if required by the instructions for authors of Nutrients;

Answer: Amended as suggested.

8. 2.1 sub-section: briefly, which type of pasta?

Answer: We reported meal composition in supplementary Table 2.

9. 2.1 sub-section: briefly, which type of salad?

Answer: We reported meal composition in supplementary Table 2.

10. 2.1 sub-section: briefly, which type of bread?

Answer: We reported meal composition in supplementary Table 2.

11. 2.1 sub-section, line 88: which type of cocoa? Which type of sugar?

Answer: cocoa was 80% from Trinidad and 20% Ivory Coast.

Sugar:100% Italian beet

12. 2.1 sub-section. They exist many types of EVOO and not the EVOO. Please, detail the type of EVOO you have used in your experiment: geographical area of production; labelled as EVOO or have you conducted analyses? Year of production and year of your experiment; monocultivar or blended (if possible detail the cultivar/s);

Answer: EVOO was provided by Monoculture (Itri area, Latina); its composition is reported separately

13. 2.1 sub-section, the authors indicate references 6, 10, 11 in which is reported a part of the experimental method. Yes, I understand, but an International reader cannot find and read 4 papers (3+ 1 actual) to understand the experimental design. Please, describe the experimental design in this manuscript (briefly if you want but insert the relevant information, mainly he ones I have listed in my comments;

Answer: Amended as suggested

14. Table 1, please use the template to prepare a table;

Answer: Amended as suggested

15. Table 1 and in the whole manuscript, when you have indicated the significance, sometime you have used P and sometime p. Please, be consistent. Anyway, here you have written P in the head of the column and it is not necessary to repeat for each line;

Answer: Amended as suggested

16. Figure 2h, and in the whole manuscript, please, separate decimals by a dot and not by a comma;

Answer: Amended as suggested

17. Figure 3, and in the whole manuscript, please, separate decimals by a dot and not by a comma;

Answer: Amended as suggested

18. Figure 4, and in the whole manuscript, please, separate decimals by a dot and not by a comma;

Answer: Amended as suggested

19. Results have to be better presented and argued;

Answer: We respectfully believe that data interpretation finds a better place in the Discussion

20. Discussion has to be widely extended also in relation with other studies conducted on this argument in light of the results obtained by the Authors. This section is too poor;

Answer: We added other references regarding the effect of EVOO on glycemic profile, that are consistent with results of the present study.

See the introduction lines 53-56

 “We and others have previously reported that in diabetic patients, intake EVOO or its component oleuropein lower post-prandial LPS and glycaemia, an effect of potentially clinical relevance as post-prandial glycemia may increase the risk of cardiovascular disease [6-9].”

 and Discussion lines 217-219

“In order to investigate the underlying mechanism, we focused on GLP-1, which is increased by EVOO administration [9,16,17] and suggested to be implicated in up-regulation of TJ proteins [23].”

21. References section is not arranged as required by Nutrients. For example, ref 6 (et al);

Answer: Amended as suggested

22. References section: sometime you have written the title of the paper in small letters (ref 11) and sometime in capital letters (ref 12), please, be consistent and use some recently published paper as a template. Small or capital letters?

Answer: Amended as suggested

23. References section. After the last page of the reference you have to write a dot and not a comma, please see you ref 13 and the whole section;

Answer: Amended as suggested

24. Please, write in blue color or evidence differently the corrections you will do also in the references section.

Answer: Amended as suggested

Round 2

Reviewer 2 Report

The authors have included all my comments.

This manuscript is a resubmission of an earlier submission. The following is a list of the peer review reports and author responses from that submission.

Round 1

Reviewer 1 Report

This manuscript focuses on an important aspect 'metabolic endotoxemia' and illustrates in a human-subject study that Extra virgin olive oil (EVOO) improves gut permeability in diabetic patients.

The authors have adhered to the guidelines in regard to conducting human trial and show that addition of EVOO to mediterranean diet or chocolate improves gut permeability and low grade endotoxemia.The experiment conducted adding EVOO to mediterannean type diet that show decrease of LPS and zonulin indicating EVOO has positive effect on permeability.

This is an important finding and needs to be researched further could prove beneficial clinically especially to diabetic patients.

Reviewer 2 Report

In this manuscript, the authors present the evaluation of extra virgin olive oil (EVOO) consumption on gut permeability-derived endotoxemia among type II diabetics and healthy subject groups. This work attempts to clarify the relationship between low-grade endotoxemia, gut permeability, and the relevant postprandial metabolic responses to EVOO consumption. The question is interesting but the paper needs major revisions as the current results is preliminary to the conclusions that the authors are trying to make.

  • Introduction: Would the authors be able to elaborate on the significance of gut permeability, EVOO intake, as well as zonulin? Cite relevant sources. The introduction does not provide enough background to suggest the novelty of this work.
  • Line 47 – 49, “We have previously reported that in T2DM intake of extra virgin olive oil (EVOO) or its component oleuropein lower post-prandial LPS and glycaemia, but it was investigated the role, if any, of gut permeability.”: Would the authors be able to clarify this sentence as it does not read well? Please provide the relevant citation to this work.
  • What is the rationale of the choice of subjects in the cross-sectional comparison? How comparable and not comparable are they in terms of other clinical parameters? It is concerning that the age range is largely different for these two groups of people so there can be many reasons why we see differences in control versus patients.
  • It is also confusing that the abstract mentioned that HS and IFG are matched for aged but clearly in the materials and methods, they are not.
  • Could the authors explain the rationale of using the test meals (chocolate and Mediterranean-type meal), with relevance to the study objectives.
  • The materials and methods needs to be written in a clear manner. There are several groups of comparison so it will be helpful when all comparisons are well described.

Reviewer 3 Report

Review of Manuscript ID: nutrients-1593188

Title: Extra virgin olive oil improves gut permeability and metabolic endotoxemia in diabetic patients

Authors: Simona Bartimoccia, Vittoria Cammisotto, Cristina Nocella, Maria Del Ben, Francesco Baratta, Pasquale Pignatelli, Lorenzo Loffredo, Francesco Violi *, Roberto Carnevale Submitted to section: Nutritional Epidemiology.

The manuscript by Bartimoccia et al., describes a collection of data analyses which aims to establish the association between consumption of extra virgin olive oil (EVOO) and markers of gut permeability. Across three different studies, the authors review 1) cross-sectional comparisons between healthy controls and those with impaired fasting glucose (IFG), as well as 2) two studies which looked at the postprandial responses (glucose, insulin, GLP-1, LPS, zonulin) of including EVOO within either a Mediterranean style meal or chocolate. There was a high degree of agreement between the findings in these three studies and these data are of interest in describing the role of a compromised gut upon chronic / acute cardiometabolic responses. That said, there are a number of areas where the clarity of the study design and the study findings could be significantly improved, and I would encourage the authors to do so prior to publication (here or elsewhere). I also have substantial reservations regarding the interpretation / clinical relevance / robustness of data that is  based on a sole postprandial assessment timepoint (at 2h) - especially when the intervention materials are energy dense and potentially fat rich which will slow gastric emptying. For a lipid based intervention, I would have expected a more prolonged observation period (0-6/8h, with frequent and repeated biological sampling so as to ascertain the kinetics of intervention material processing.

I have separated my comments by manuscript section.

Title:

  • Despite the title stating ‘in diabetic patients’, all of the data reported appears to be in patients with impaired fasting glucose, not diabetes. Perhaps these participants can be described as having pre-diabetes. Likewise, the title states ‘Extra Virgin Oil’, yet one of the intervention studies appears to be enriching chocolate with oleuropein, not EVOO per se.. the title shoudl reflect this more accurately.
  • Replacing the word ‘improves’ with ‘reduces’ would help the reader to understand the directionality of the study findings.

Abstract:

  • Ln 17, replace ‘via’ with ‘by reducing’.
  • Ln 18 – abbreviate lipopolysaccharides
  • Can the authors find a better way to make it clear to the reader that the data presented is from 3 separate studies which are not amalgamated? These separate studies need labelling in a way (throughout the manuscript) which improves the clarity for the reader – perhaps differentiating the studies with a numerical or alphabetical label.
  • Critical details of the types of study included are missing and should be more apparent e.g. a) the words; ‘cross-sectional / observational’ should be used for study 1; b) study 2 gave a single Mediterranean meal (it would be good to confirm what was in it), not a Mediterranean diet and it was a randomized, ?parallel? trial(this is unclear in the manuscript), c) study 3 gave oleuropein-enriched chocolate, not EVOO, and the authors should confirm either what the level of oleuropein was, or the equivalent amount of EVOO you would find it in, also what sort of chocolate it was – the detail later in the manuscript says chocolate spread.. but it’s not clear if this is white, milk, dark chocolate and what the other polyphenolic constituents are of the chocolate.
  • Currently, it’s very difficult for the reader to appreciate which of the three studies the authors are describing in the results section – it would help if this was made clearer for example – ‘in study 1, cross-sectional data showed…’ ‘when a Mediterranean meal was supplemented with EVOO, we observed…’ etc.
  • Please amend the conclusion to be consistent with the above comments (i.e. oleuropein provision, not EVOO, and the intervention with a Mediterranean meal, not a diet) and also to make it apparent that your endpoints are ‘markers of gut permeability

Keywords:

  • should this include oleuropein, Impaired fasting glucose (IFG)
  • Should type 2 diabetes (T2DM) be removed.

Introduction:

  • Ln 47-49: the authors are confirming that the role of gut permeability in lower LPS and glycemia was not previously explored, but the sentence isn’t quite correct – it needs a ‘not’ somewhere in it.
  • My understanding of the study design was to assess markers of gut permeability on patients with IFG, not T2DM patients.. people can have IFG, but not T2DM.

Materials and Methods:

  • The description of the two RCTs is inadequate. It is not clear or consistent what the characteristics are of the participants involved. Whilst we know that 10g of EVOO was added to a meal (which we know was 894kcal), we don’t know what the enrichment level was in 40g of ‘chocolate’ or the energy intake level etc... which we later find out is chocolate spread. Similarly, whilst we can identify that the chocolate study is a single blind, crossover study – are we to assume that the ‘meal’ study is a randomized, parallel designed study? Were there any dietary restrictions or important preparatory steps in place prior to either of the 2 RCTs which will help us attribute effects to the intervention?
  • It would be very helpful to have a table (even if it was supplemental) which clearly showed the nutritional composition of the intervention materials – both in terms of EVOO or it’s component parts, other polyphenols what might also be expected to affect postprandial responses, and also macronutrient composition between intervention arms – especially fat content, which is likely to affect gut motility and ADME profiles.
  • Can the authors confirm how the disparity in energy or fat intake in the ‘control’ arms was accounted for? Did the authors add a different, non-EVOO oil? I appreciate some of these data may be in previous publications, but it is frustrating to be unawares of key characteristics such as age, sex, BMI, amount of active intervention ingredient etc. which can significantly affect interpretation and comparison between studies.
  • Can the authors confirm the clinical trial registration details for the human studies so that it is transparent to the reader what the aims, objectives, intervention arms were.
  • Are there any details of the %CV for the ELISA analyses for glucose, insulin and GLP-1 that were performed (so that these data reports are consistent with the data presented for LPS and zonulin) – were the assessments done in triplicate, duplicate, singletons? Did the plates contain any controls or internal standards?   

Results:

  • Can the authors include some baseline data information for the 3 studies – i.e. usually table 1 would include participant characteristics.
  • Can the authors please signpost the results better – i.e. lines 107-109 appear to relate to the first study… can this be stated? If it’s made clear, then the reader may assume that you are assessing the levels of these biomarkers across all 3 studies at baseline (which I don’t think you are). The same comment can be made for the results that seem to be derived separately from each of studies 2 and 3.
  • Can Figures include the study number that the results relate to?
  • Can there be better consistency with the figure labels – T0 and T2h sometimes appear slanted for no obvious reason (which make it difficult to read), there is a T1 on figure 2d), is there a reason why the 2 has an ‘h’ but the 0 does not?
  • Can the authors include the participant numbers in each of the groups on the figures; i.e. IFG=30, HS= … this can go as a footnote if it keeps the figures clearer.

Discussion:

  • Can the authors comment on the validity of assessing postprandial meal / fat-based assessment over a 0-2h period? In each of the postprandial slides, there is no return to baseline levels so we are only seeing a single reference point after feeding to try and interpret what is going on – a more robust postprandial assessment would include sufficient measures to look at AUC, or at least until baseline level was reached. This is a clear limitation and needs raising in the discussion as I suspect there is not the data to be able to add this to the manuscript.
  • Because it is unclear from the description how the control arms matched fat and / or energy content with the EVOO based intervention arms, there remains the possibility that the observations are driven by a slowing of gut motility and gastric emptying i.e. non-EVOO may have quicker gastric emptying, which precipitates higher responses, which last for a shorter time; whereas EVOO may have slower gastric emptying *(due to higher fat content), which may show up as lower responses at 2h, but these may be extended for a much longer time (as a consequence of slower gastric emptying) which, when reviewed as AUC, may not show the same significant difference that we are seeing at just 2h.
  • Can the authors include something in the discussion about the clinical importance of their results (which are based on a quite transient period of time)? It would be nice to see a discussion point about the suppression of LPS and Zonulin still being substantially higher than that observed for HS and to what extent the levels would need to fall further to observe sustained health improvements.
  • From other studies, can the authors reassure the readers that the postprandial response captured in this observation period (0-2h) is sufficient to draw conclusions about fatty acid ADME and it’s transient effect on markers of metabolism and interaction with the gut